# Tensile force-induced cytoskeletal remodeling: Mechanics before chemistry

**Xiaona Li**[1‡], **Qin Ni**[2‡], **Xiuxiu He**[1], **Jun Kong**[1], **Soon-Mi Lim**[3¤], **Garegin A. Papoian**[4], **Jerome P. Trzeciakowski**[3], **Andreea Trache**[3,5], **Yi Jiang**[1]*

**1** Department of Mathematics and Statistics, Georgia State University, Atlanta, Georgia, United States of America, **2** Department of Chemical & Biomolecular Engineering, University of Maryland, College Park, Maryland, United States of America, **3** Department of Medical Physiology, Texas A&M University Health Science Center, Bryan, Texas, United States of America, **4** Department of Chemistry & Biochemistry, University of Maryland, College Park, Maryland, United States of America, **5** Department of Biomedical Engineering, Texas A&M University, College Station, Texas, United States of America

¤ Current address: Department of Chemistry, Texas A&M University, College Station, Texas, United States of America
‡ These authors share first authorship on this work
* yjiang12@gsu.edu

**Data Availability Statement:** All relevant data are within the manuscript and its Supporting Information files. The original appeared in https://www.jove.com/video/2072/live-cell-response-to-mechanical-stimulation-studied-integrated

## Abstract

Understanding cellular remodeling in response to mechanical stimuli is a critical step in elucidating mechanical activation of biochemical signaling pathways. Experimental evidence indicates that external stress-induced subcellular adaptation is accomplished through dynamic cytoskeletal reorganization. To study the interactions between subcellular structures involved in transducing mechanical signals, we combined experimental data and computational simulations to evaluate real-time mechanical adaptation of the actin cytoskeletal network. Actin cytoskeleton was imaged at the same time as an external tensile force was applied to live vascular smooth muscle cells using a fibronectin-functionalized atomic force microscope probe. Moreover, we performed computational simulations of active cytoskeletal networks under an external tensile force. The experimental data and simulation results suggest that mechanical structural adaptation occurs before chemical adaptation during filament bundle formation: actin filaments first align in the direction of the external force by initializing anisotropic filament orientations, then the chemical evolution of the network follows the anisotropic structures to further develop the bundle-like geometry. Our findings present an alternative two-step explanation for the formation of actin bundles due to mechanical stimulation and provide new insights into the mechanism of mechanotransduction.

## Author summary

Remodeling the cytoskeletal network in response to external force is key to cellular mechanotransduction. Despite much focus on cytoskeletal remodeling in recent years, a comprehensive understanding of actin remodeling in real-time in cells under mechanical stimuli is still lacking. We integrated tensile stress-induced 3D actin remodeling and 3D

**Funding:** The work was supported in part by Public Health Service grants R01CA201340 and 1R01EY028450 from the NIH/NCI and NIH/NEI, respectively to Y.J., K25CA181503 and U01CA242936 from NIH/NCI to JK, National Science Foundation grant CHE-1800418 and PHY-1806903 to GP. The experimental work was supported by NSF CAREER 0747334 and AHA-National SDG 0835205N to AT. The funders had no role in study design, data collection and analysis, decision to publish, or preparation of the manuscript.

**Competing interests:** The authors have declared that no competing interests exist.

computational simulations of actin cytoskeleton to study how the actin cytoskeleton form bundles and how these bundles evolve over time upon external tensile stress. We found that actin network remodels through a two-step process in which rapid alignment of actin filaments is followed by slower actin bundling. Based on these results, we propose a "mechanics before chemistry" model of actin cytoskeleton remodeling under external tensile force.

## Introduction

Cells adapt to local mechanical stresses by converting mechanical stimuli into chemical activities that alter the cellular structure-function relationship and lead to specific responses [1–3]. Cellular response to mechanical stimulation is a balance between contractile elements of the cytoskeleton, cell-matrix adhesions, and extracellular matrix [4]. Although cellular mechano-transduction has been an active field of research for a number of years, the process by which transduction of external mechanical signals across the cellular cytoplasm induce cytoskeletal remodeling is not well understood. The most important question in the field of mechanobiology is '*how do cells sense and integrate mechanical forces at the molecular level to produce coordinated responses necessary to make decisions that change their homeostatic state?*'

Vascular smooth muscle cells (VSMCs) provide an excellent model system to study the mechanotransduction process. The mechanism by which VSMCs sense and adapt to external mechanical forces that result in cytoskeletal remodeling (6–8) is critical for understanding arterial disease pathology. *In vivo*, they sense and respond to mechanical forces generated by pulsatile blood pressure changes through alteration of signal transduction pathways to induce remodeling of their cytoskeleton and adhesions [5, 6]. Thus, VSMCs residing in the vessel wall are mainly subjected to circumferential stretch and axial stress [7–9]. Circumferential stretch generated by the pulsatile blood flow exerts dynamical mechanical stimulation on the vessel wall in a direction perpendicular to the direction of blood flow. This is a well-recognized mechanical stressor and its biomechanical effects were well studied [10, 11]. Axial stress in the vessel wall arises from longitudinal loading along the vessel length [12]. Even though axial stress (i.e., tensile force) has been known as an important mechanical stressor of the vessel wall for a long time [13, 14] and a fundamental contributor to vessel wall homeostasis (12), less attention was given to studying its biomechanical effects at the cellular level.

In anchorage-dependent cells, external mechanical forces are imposed on a pre-existing balanced force equilibrium generated by cytoskeletal tension [15–17]. Thus, forces acting on a cell will induce cytoskeleton deformation throughout the cell, such that the actin cytoskeleton remodels to better sustain the external load. Actin cytoskeleton consists of semi-flexible actin filaments, myosin motors, and crosslinking proteins. It has been proposed that *de novo* actin polymerization is critical for actin fiber formation in migrating cells [18], while the aggregation of existing actin filament fragments is most likely for stationary cells in a static environment [19]. Mechanical stimulation of stationary VSMCs in tissue represents an intermediate state, in which cells must dynamically adapt to their native, mechanically active environment. It is not known which mechanism is favored in this normal functional homeostatic state. Moreover, research has shown that cells adapt to external force by activating mechanically-sensitive signaling pathways that involve conformational changes of proteins at cell-matrix adhesions (e.g., integrins, vinculin, talin, etc), and promote actin filament polymerization [20].

Our previous experiments on VSMCs suggested that cellular adaptation to the applied tensile force is a characteristic of the integrated cell system as a whole [21]. To address how application of external tensile force induces actin cytoskeleton remodeling, we combined imaging techniques with simultaneous mechanical stimulation of single cells using fibronectin-functionalized atomic force microscope (AFM) probes [22]. Thus, we found that mechanical stimulation not only increases alignment of actin filaments, but also induces actin bundling measured by increased fluorescence intensity of F-actin [23].

Here, we build upon these experimental results and investigate the biomechanical effects of axial stress at cellular level using computational modeling, by asking how tensile force induces actin cytoskeleton adaptive remodeling? During the adaptation process, the actin cytoskeleton remodels to better sustain the external load [24–26]. Thus, actomyosin networks crosslinked by α-actinin and other crosslinking proteins are able to adapt to external forces via *fast* mechanical response, in which stress relaxation occurs on the timescale of seconds [27–30]. However, cytoskeletal reactions, such as actin (de)polymerization or myosin II activation that continuously converts chemical energy into mechanical force, remodel the actomyosin networks on a *slower* pace, on a time scale of minutes [31–33]. As a result of myosin dominant mechanochemical dynamics, actomyosin networks tend to contract [34, 35]. Prior computational models have investigated remodeling of the actin cytoskeleton due to slower chemical reactions [36–40], however, how external mechanical stimuli induce the active formation of actin bundles is still poorly understood.

To better understand the detailed spatiotemporal dynamics of cytoskeletal reorganization due to external mechanical loading, we simulated the mechanical and chemical dynamics of the actin cytoskeleton using the MEDYAN (MEchanochemical DYnamics of Active Network) software [41]. In our simulations, we model the active cytoskeletal networks using polymer mechanics of semi-flexible filaments, crosslinking proteins, and motor proteins. A stochastic reaction-diffusion scheme was used to simulate chemical reactions, including myosin activation, crosslinking protein binding, and actin filament assembly. Additionally, we have applied external tensile forces to the actin network to mimic the AFM mechanical stimulation in the experiments. In these systems, a few filaments were initially anchored to a simulated AFM probe, in addition to a free filament pool. The external force was applied by moving the simulated AFM probe upward, by increasing the amplitude of z-axis displacement. In highly crosslinked actomyosin networks, the external force exerted on a small fraction of filaments would transmit to the entire system that changes their homeostatic state in microseconds [42]; this will be considered as the *fast* mechanical response. After each tensile force was applied, the system was allowed to evolve for minutes, such that we were able to study how the actin network adapts under a slower chemical response.

Both experiments and simulations suggest that the external tensile force applied on actin networks quickly induces alignment of actin filaments along the direction of force, and this directional alignment is independent of longer timescale chemical response. In addition, the formation of actin bundles as a result of external tensile force relies on both the faster mechanical response and the slower chemical response. We hypothesized that cellular cytoskeletal adaptation to external tensile forces and formation of actin bundles follows a "mechanics before chemistry" process.

## Results

### Actin cytoskeleton reorganization in live VSMCs under mechanical stimulation reveals a two-step adaptive response

Live VSMCs expressing mRFP1-actin-7 were subjected to the mechanical loading delivered by the AFM probe at the apical cell surface (Fig 1A). Vertical forces (along the z-axis) applied

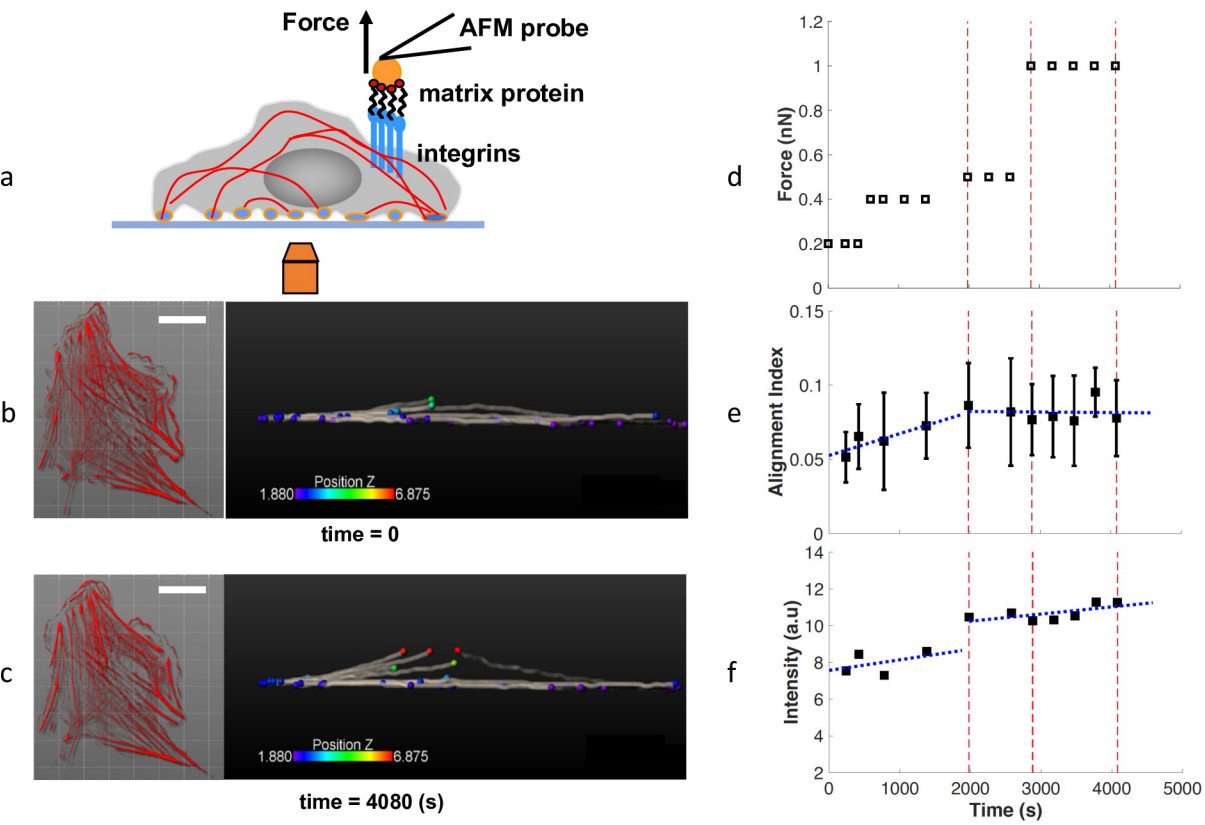

**Fig 1. Response of VSMC to external pulling force.** (a) Schematic of a VSMC mechanically stimulated with a FN functionalized AFM probe and simultaneously imaged by spinning-disk confocal microscopy. (b-c) Fluorescence images of VSMC expressing mRFP1-actin-7 (left) and the 3D reconstructed image of the representative segmentation of actin filament bundles (right) for before (b) the AFM probe displacement at time 0 min, and after (c) the AFM probe displacement at time 68 min. Scale bar: 20 μm. Left panels used with permission from JOVE [21]. (d) The scheduled pulling force in three phases: small, intermediate and large forces. (e) The alignment index for the actin filament bundles increased rapidly as small force was applied, but slowed down as the force increased. (f) The normalized intensity for mRFP1-actin-7 increased steadily through all force ranges. Blue lines: piece-wise linear fit for forces below 0.5 nN and ≥0.5 nN.

through a fibronectin (FN) functionalized probe induced cytoskeletal remodeling by pulling on cortical actin through a FN-integrin-actin linkage [9, 21]. Cell responses to the probe displacement over time were recorded using spinning-disk confocal microscopy (S1 Video). The reconstructed 3D-images of the actin cytoskeleton were used to segment actin bundles in 3D (Fig 1B and 1C). We used these 3D bundles to calculate an average fiber alignment index in the direction of the pulling force. The alignment index is defined as the average of $cos(\theta)$, where $\theta$ is the acute angle between each filament segment and the direction of the force (Z-axis) (Eq 1 in Methods). With a value between 0 and 1, the alignment index equals to 1 for perfect alignment with the Z-axis, and 0 for alignment perpendicular to the Z-axis. The alignment index increases right after the application of an external force, but levels off (Fig 1E) upon larger pulling forces. Note that the small alignment index value is due to the cell aspect since the VSMCs lay flat on the substrate, and the majority of the filaments were perpendicular to the Z-axis. In addition, the normalized fluorescence intensity of actin filaments increased steadily as the AFM displacement continued (Fig 1F). These experimental results show a force-induced actin cytoskeleton remodeling via the directional alignment and actin fiber bundling.

## Rapid formation of actin bundles in response to tensile force in MEDYAN simulations

To understand the molecular mechanisms of the actin cytoskeleton reorganization under tensile force application using the AFM probes, we designed computational simulations of actin networks with external tensile force using MEDYAN software. We generated 300 free filaments in a 3×3×1.25 μm³ simulation box, initially as a random network, and another 30 filaments attached to an AFM probe located at the center of the upper boundary of the simulation box. The number of filaments attached to the AFM-probe was chosen based on the reported number of filaments in actin bundles [19]. The simulation box contained 20 μM of actin, 2 μM of non-muscle myosin II (NMII), and 2 μM of α-actinin crosslinkers. The simulated AFM probe was displaced by a distance $d$, every 150 seconds. Each pull (Z-axis tensile force application) created a 250 nm or 500 nm step displacement of the simulated-AFM probe, generating tensile force on the filaments attached to the probe via stiff harmonic springs (Fig 2A). The amplitude of step displacement size $d$ is linearly proportional to the pulling force of the AFM probe. Chemical interactions, including filaments treadmilling, myosin activation, and α-actinin crosslinking, took place throughout the simulations. We varied the pulling patterns (Fig 2B) to simulate the different pulling forces in the experiment (Fig 1D).

Interestingly, pulling on only a small fraction of filaments attached to the AFM-prove is sufficient to alter the actin filament structure of the entire network. After 900s and five AFM probe pulling steps, each with $d$ = 500 nm (case i), the actin networks reorganized into a bundle (Fig 3A and S2 Video), which is approximately 2 μm long and around 500 nm thick. These actin bundles have mixed filament polarity, i.e., plus ends or minus ends of filaments are randomly distributed (S1 Fig in Supporting Information). In contrast, actin networks free of external force geometrically collapsed into a globular cluster-like structure (Fig 3B and S3 Video), as a result of contractility driven by myosin motors and crosslinkers. Reducing the step size $d$ in Cases ii and iii creates an intermediate geometry between the bundle and cluster

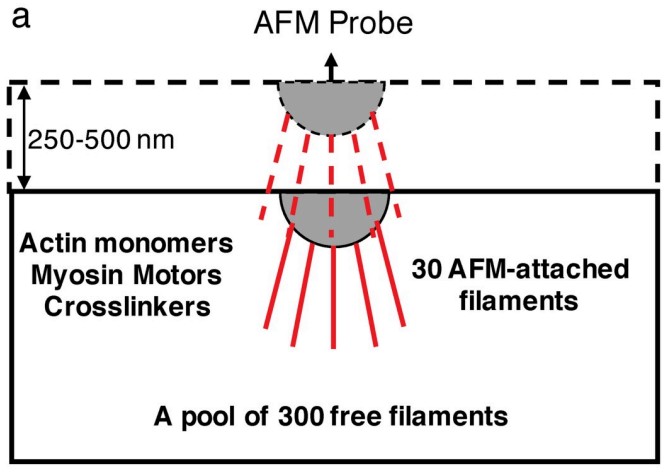

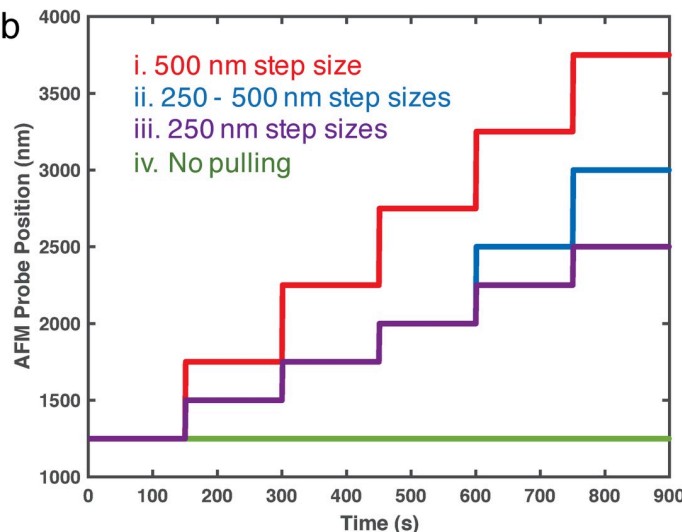

**Fig 2.** (a) A sketch of the simulation setup. The simulation box is 3 μm in x and y directions, and the initial height (z-direction) is 1.25 μm. The simulation box contains 300 free actin filaments, as well as diffusible G-actin, myosin, and α-actinin linkers. A semi-spherical AFM probe is located at the upper boundary, and 30 filaments are attached to the probe via stiff harmonic springs. At the beginning of simulations, all filaments are 0.108 μm long (containing 40 actin subunits). The input G-actin concentration is much higher than the equilibrium concentration, making actin filaments rapidly elongate. An average length of 0.8 μm is achieved and maintained after around 40s of simulation. (b) Simulated AFM-probe position, equivalent to the height of upper boundary, as a function of time for Cases i-iii. The control case (Case iv) is without AFM probe and without filament attachment, with only the upper boundary moving in the same way as in Case i to avoid potential boundary effects.

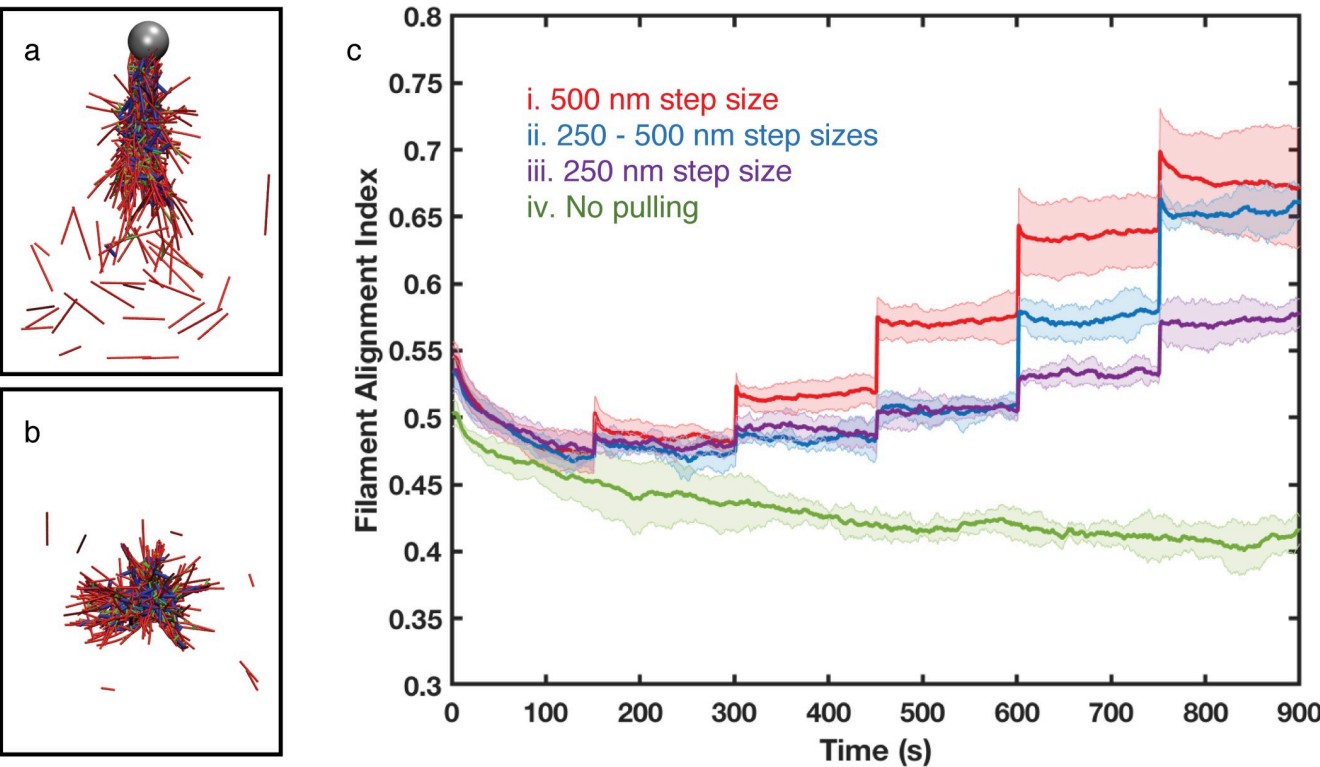

**Fig 3.** (a-b) Representative snapshots of (a) bundle-like actin networks under Case i pulling condition at time t = 900 s, and (b) cluster-like actin networks without external force at time t = 900 s. Actin filaments, myosin motors, and crosslinkers are shown in red, blue, and green cylinders, respectively. The gray sphere in (a) represents the AFM probe. (c) Filament alignment index along the Z-axis for 500 nm step size (red, Case i), mixed step sizes (250 nm for the first three pulling events and 500 nm for the last two, blue), 250 nm step size (purple, Case ii) and no AFM-probe pulling (green, Case iii). α-actinin linker: actin is 0.1 and myosin:actin is 0.005 in all simulations. Error bars represent the standard deviation from the mean in 5–10 simulation replicates.

(S2 Fig). If the step size is further reduced to 0, but the 30 filaments are maintained attached to the simulated AFM probe, the geometry would be similar to the cluster (S3 Fig). Moreover, if we release the filaments from the simulated AFM probe after bundle formation, then actin bundles would also tend to collapse into globular clusters (S4 Video).

The actin bundle formation was also regulated by the number of actin filaments attached to the AFM-probe. When too few filaments were attached to the AFM probe, the pulling force was insufficient to generate a bundle (S4A Fig). In some of the simulations, after pulling, the actin filaments that were attached to the AFM-probe would disconnect completely from the free actin filament pool. For the myosin motor and α-actinin concentrations used in our simulations, we found that about 20 actin filaments need to be attached to the probe for actin bundle formation. On the other hand, increasing the number of AFM-probe attached filaments lowered the density of the free filaments. As a consequence, most free filaments could only collapse into small globular clusters locally and were unable to join the actin bundle formed by the filaments attached to the AFM-probe (S4B Fig).

## Tensile force induces actin alignment in MEDYAN simulations

To investigate actin filament alignment during actin bundle formation, we calculated the alignment index $cos(\theta)$ as described in the experimental section. The alignment index increases immediately after each of the AFM-probe pulling events in all three pulling patterns tested (Fig 3C, Case i-iii). In the simulation, mechanical equilibration is instant, therefore these rapid

jumps suggest very strong mechanical responses. Moreover, the directional filament alignment is regulated by the magnitude of the external tensile force, since reducing the pulling step size amplitude (compared to Cases ii and iii) results in a weaker alignment response. On the other hand, the directional alignment barely changes at long timescale in all step size patterns. Since the long timescale response is regulated by slower chemical evolutions, we hypothesize that the directional alignment of actin filaments in response to tensile force is primarily due to fast mechanical adaptation.

## Two-step development of actin bundles depends on both faster mechanical alignment and slower chemical response

To further analyze the formation and evolution of actin bundles, we next defined a cylinder-shaped boundary under the AFM probe (500 nm in diameter). More than 80% of the total F-actin accumulated within this boundary towards the end of simulations under the Case i pulling condition, suggesting that monitoring F-actin accumulation in the bundle region provides a simple but robust way to quantify the bundle development process. We observed instant F-actin accumulation after each pulling event (Fig 4A), while reducing step size hindered the accumulation (S2A Fig). Similar to the directional alignment, these results suggest that actin bundle development relies on the fast, mechanical response.

Surprisingly, the accumulation of actin filaments into the bundle kept increasing steadily between pulling events, suggesting that slower chemical dynamics contribute to bundle development as a result of the adaptation to force. To capture the long timescale of F-actin recruitment, we calculated the F-actin recruitment rate in the defined bundle region (Fig 4B). The control case without external pulling (Case iv, green line) shows the chemically driven F-actin recruitment, as a result of myosin and α-actinin induced contractility and bundling, respectively. Similarly, the recruitment rate of F-actin during the intervals between pulling (Case i, red line) is always positive, showing net recruitment of F-actin. The rate of F-actin recruitment for bundling is lower than that for actin clustering into globular foci in the control case.

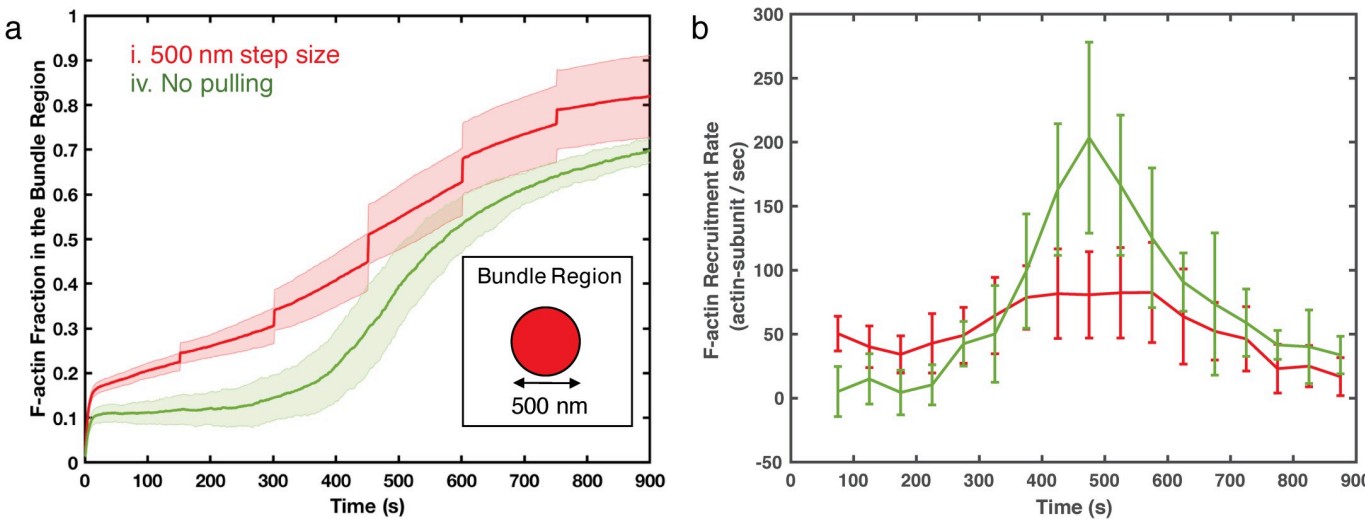

**Fig 4.** (a) The F-actin fraction in the bundle region as a function of time for actin networks under 500 nm displacement steps (red, Case i) and actin networks free of external force (green, Case iv). The bundle region is defined as the volume under the simulated AFM probe, which is a cylindrical region of 500 nm in diameter and the height of the simulation box. The box size varies over time based on the position of the AFM probe. Insert shows a 2D illustration of the bundle region. (b) The rate of F-actin accumulation in the bundle region from simulations with the AFM probe pulling force (red, Case i) and without AFM probe pulling- force (green, control Case iv). The recruitment rates are calculated by linear-fitting of the data points every 50 seconds. Shaded colors and error bars are the standard deviations of 10 replica simulations for Case i and 5 replicas for Case iv, respectively.

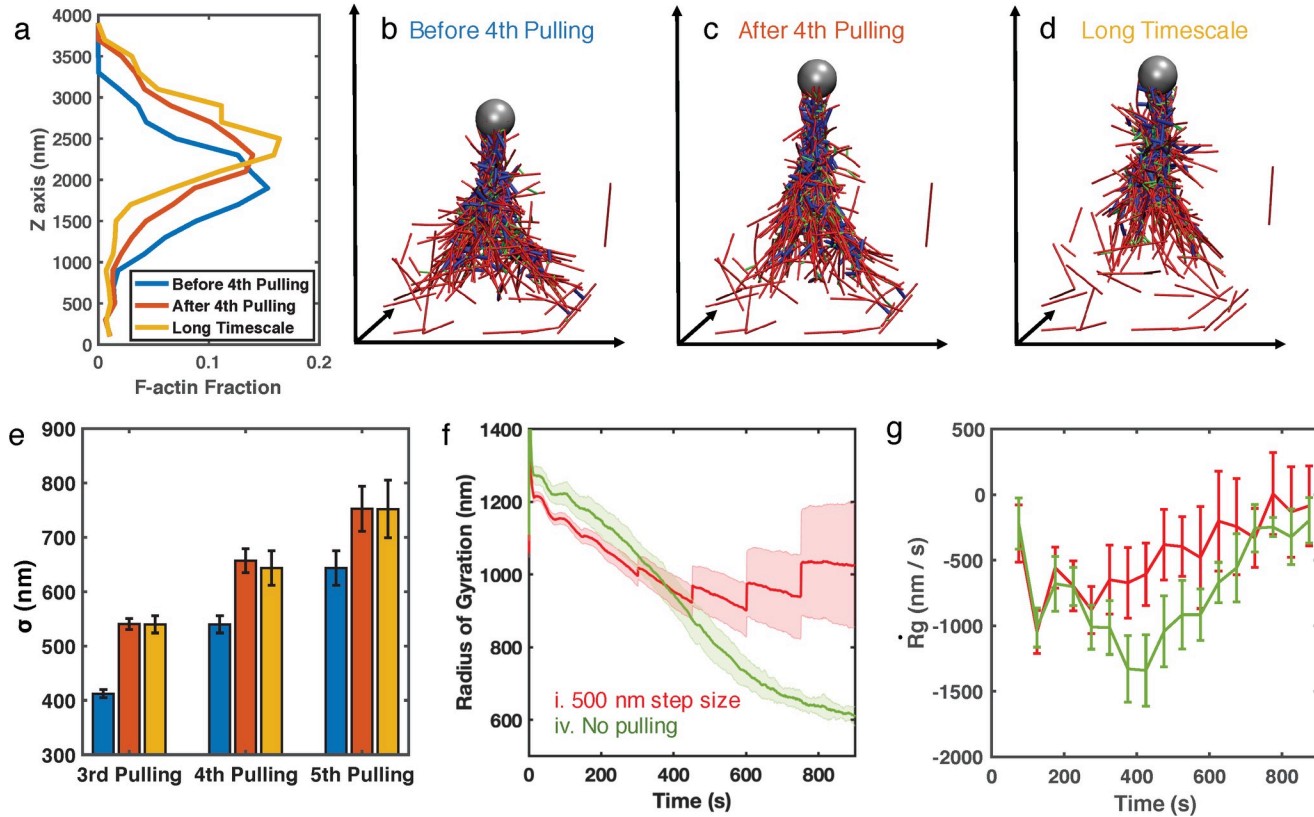

**Fig 5.** (a) F-actin distribution along the force direction (Z-axis) of the most representative trajectory before the 4[th] pulling at t = 600 s (blue), after the 4[th] pulling at t = 601 s (orange), and after the long-timescale chemical evolution at t = 750 s (yellow). (b-d) Corresponding simulation snapshots before pulling, after pulling, and after chemical evolution, respectively. (e) Standard deviations (σ) of the F-actin distribution along the force direction before pulling (blue), after pulling (orange), and after 150 s of chemical evolution (yellow) at the third, fourth, and fifth pulling events. σ are averaged over 10 duplicated trajectories, and error bars represent the standard errors. (f) The radius of gyration, Rg and (g) the rate of Rg change, $\dot{R_g}$, of actin networks in Case i with 500 nm pulling (red) and in Case iv without pulling (green). Shaded colors and error bars are the standard deviations of 10 duplicated trajectories for Case i and 5 duplicated trajectories for Case iv.

To further explore the significance of chemical evolution of bundle development, we tested three different conditions with "insufficient" chemical evolution. First, we reduced the myosin concentration from 2 μM to 0.4 μM. Without sufficient myosin, the network was unable to generate enough contractility of the actomyosin network, leading to high actin filament dispersion (S5 Video). Second, by reducing α-actinin crosslinker concentration from 2 μM to 0.4 μM, the F-actin network could not form properly (S6 Video). Although myosin motors still generated contractility, the actin fiber network is fragmented as disconnected actin foci. Lastly, we shorten the time between each pulling from 150 seconds to 10 seconds. Only a small fraction of actin filaments bundled together and followed the upward movement of the simulated AFM probe, disconnecting from the rest of the filaments (S7 Video).

F-actin distribution further showed that the tensile force application using AFM-probe immediately stretches the actin fiber network along the direction of force (Fig 5A–5C), leading to a wider distribution. As a result, the standard deviations (σ) of these distributions increased right after pulling (Fig 5E). When we measure the radius of gyration (Rg) to quantify the cluster size of actin networks, we also find instantaneous jumps similar to those in the filament alignment and recruitment results (Fig 5F). These instant stretches eventually shape actin networks into thinner actin bundles. Furthermore, these actin bundles maintain their geometric

structures at a longer timescale. The F-actin distribution of actin bundle networks shifts slightly towards the force direction after 150 seconds of chemical evolution (Fig 5A), but the shape and the standard deviation from the mean, σ, remain almost the same (Fig 5D and 5E). In addition, the contraction rate, measured as the rate of Rg change ($\dot{R}_g = \Delta R_g \big/ \Delta t$), is much slower than that for the actin globular clusters in the control case without force application (Fig 5G). These observations are consistent with the slower F-actin accumulation rate in the bundle region, as shown in Fig 4B, suggesting that the actin bundle structure is more stable than the actin cluster. These results are also in agreement with the fact that the actin bundle can preserve its shape and would not contract into clusters under myosin driven contractility at longer timescale.

## Discussions

Mechanotransduction is the process by which cells convert mechanical stimuli into biochemical activity. A key aspect of the mechanotransduction is that cells remodel their cytoskeleton in response to mechanical stimuli. To study external force-induced adaption of the actin cytoskeleton, AFM was used to apply external, tensile forces on single cells adherent on a substrate. Cell responses measured through probe displacement over time are directly dependent on the intrinsic contractility that modulates the function of the actomyosin apparatus. The observed rapid rise in actin fiber alignment upon tensile force stimulation contrasts with the continuous growth of actin fluorescence intensity, leading to our hypothesis of 'mechanics before chemistry': fast mechanical stimulation-induced actin bundle alignment, followed by a slower chemical driven process to stabilize the actin bundle structure.

To explore this hypothesis, we developed a new feature in the MEDYAN software that mimics the conditions of our AFM mechanical stimulation experiments. Our simulation results reveal that tensile force triggers a rapid mechanical adaptation of actin networks that induces actin filament to align along the direction of force application, and promotes actin bundle formation. We also found that slower chemical evolution is essential to the formation of actin bundle, which requires the integration of actin networks through α-actinin crosslinking followed by myosin activation and eventual further actin recruitment to the bundle. Moreover, we found that actin bundles generated in our simulations are stable since they contract much slower than networks free of external force.

Thus, our simulations agree with the experiments, supporting a "mechanics before chemistry" hypothesis as an alternative two-step explanation regarding how active cytoskeletal networks adapt to external mechanical stimuli in real-time. In the control case of actin networks without external forces, actomyosin network contraction does not have a bias towards a specific direction, leading to an isotopic collapse into globular actin clusters (Fig 6A). The external tensile force first stretches the actin cytoskeletal network, forcing filaments to align, as a rapid mechanical response, which initializes anisotropic actin bundle-like structures. Longer time scale chemical processes further stabilize the actin bundle structures that can preserve the anisotropy (Fig 6B). As a result, the contractility generated by subsequent chemical evolution follows the anisotropic distribution, which strengthens actin bundles by recruiting more actin filaments while maintaining the bundle shape.

Actin cytoskeleton plays a crucial role in maintaining cellular shape and supporting force transmission to and from extracellular substrates. Numerous studies have demonstrated the direct coupling between mechanical forces and chemical signaling. Mechanical stretch alters the conformation of integrins [43] such that their cytoplasmic β-tails provide binding sites for focal adhesion proteins (43) and further induce assembly of an adhesion complex at the site of force application [44, 45]. This process is followed by actin stress fiber remodeling, which is

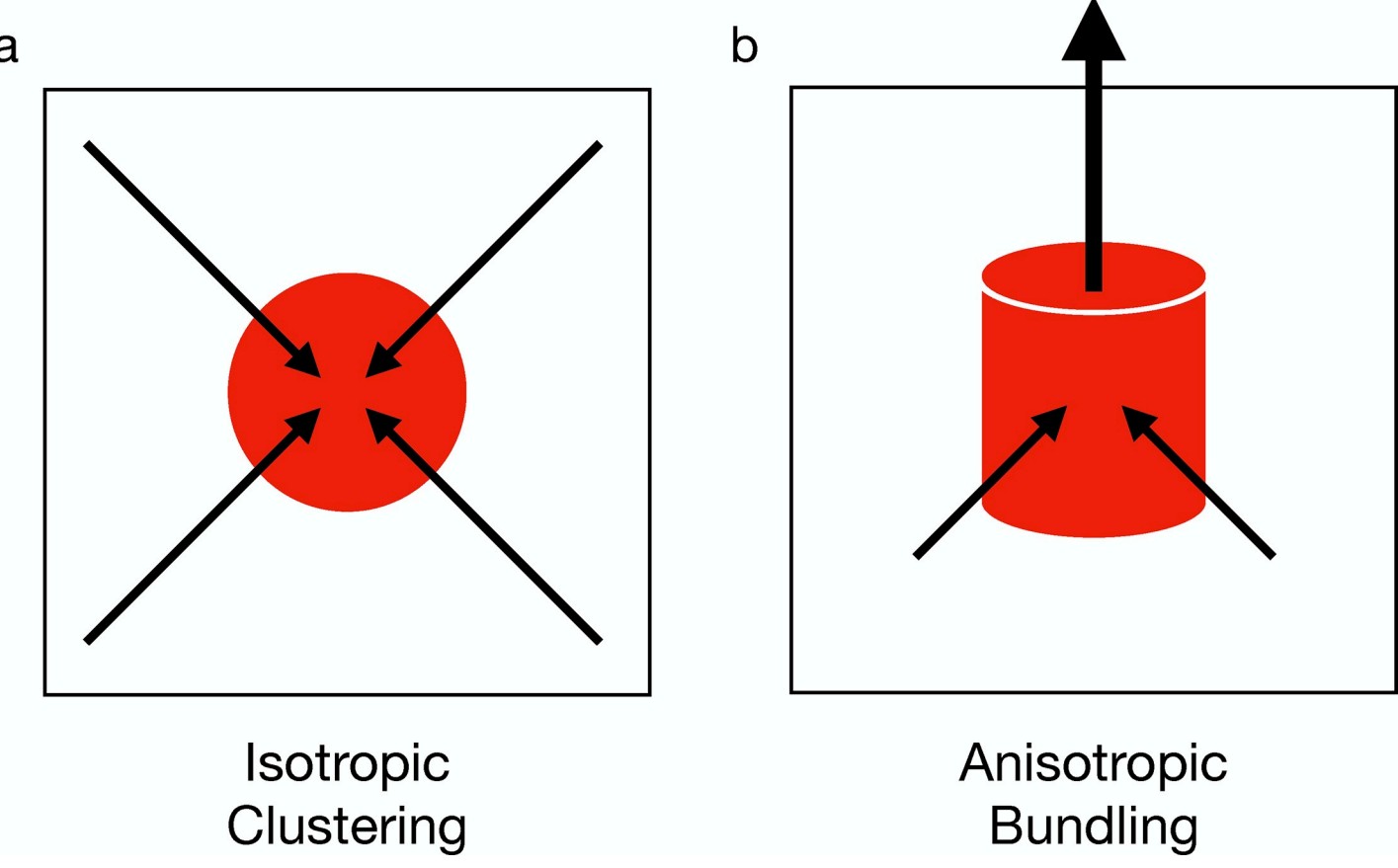

**Fig 6. Motor-driven chemical evolution generates contractility that induces the geometric collapse of the actin network.** In random networks without external forces, the geometric collapse would be isotropic, causing filaments to cluster into globular foci (a). However, the external tensile force induces filament directional alignment and favors anisotropic chemical evolution, resulting in filament bundling (b).

necessary to redistribute physical forces needed for cell contraction and to enable cell adaptation to the extracellular microenvironment [46, 47]. Moreover, sensing of substrate stiffness via integrins further triggers the adaptation of cellular cytoskeleton in less than 100 ms [48], proposing a 'mechanics first' mechanism of cellular response that supports our hypothesis. Thus, when the cell experiences an external force, the cytoskeletal adaptation will first elicit the actin fiber rearrangements (mechanical) before spending ATP to initiate the chemical reactions (chemical).

In summary, we integrated *in vitro* cellular biophysical experiments with *in silico* modeling to investigate the effects of external load on the actin cytoskeleton network. Our experimental data and modeling results suggest that under tensile force actin filaments align first, and then contractility induced by chemical evolution takes place to further restructure the cytoskeleton. The mechanical stimulation of stationary cells (*in vitro* or in tissue) represents an intermediary state of dynamic adaptation to stress of stationary cells placed in a mechanically active environment (i.e., vessel wall). Thus, our results suggest that in this intermediate cellular state, short timescale mechanical structural adaptation operates before chemical evolution necessary to further remodel the actin network. This study lays the groundwork for further studies related to predicting cellular adaptation to mechanical stimulation, which will be important for understanding diseases that involve changes of cellular stiffness, e.g., in cancer, hypertension and aging.

## Methods

### Experimental methods

**Vascular smooth muscle cell cultures and transient transfections.** VSMC were previously isolated from rat cremaster arterioles [49] and handled as previously described [23]. Briefly, cells were cultured in a smooth muscle cell culture media containing Dulbecco's Modified Eagle Medium (DMEM) supplemented with 10% fetal bovine serum (FBS), 10 mM HEPES (Sigma, St. Louis, MO), 2 mM L-glutamine, 1 mM sodium pyruvate, 100 U/ml penicillin, 100 μg/ml streptomycin and 0.25 μg/ml amphotericin B. Cells were trypsinized and transient transfections were performed according to manufacturer's protocol by using the Nucleofector apparatus (Lonza, formerly Amaxa Biosystems, Gaithersburg, MD) with Nucleofector kit VPI-1004. Then, cells expressing mRFP1-actin-7 were plated on 60 mm MatTek glass bottom dishes (Ashland, MA, USA) in phenol-red free cell culture media, and incubated overnight in 5% CO2 at 37 °C. The plasmid mRFP1-Actin-7 was a gift from Michael Davidson (Florida State University, Tallahassee, FL). Unless otherwise specified, all reagents were purchased from Invitrogen (Carlsbad, CA, USA).

**Vascular smooth muscle cell imaging.** The integrated microscope system used for these studies was described in detail (45). Briefly, the system was constructed using an inverted Olympus IX-81 microscope (Olympus Corp., NY). An atomic force microscope (XZ Hybrid Head, Bruker Instruments, Santa Barbara, CA) was set on top of the inverted microscope and a Yokogawa CSU 22 spinning-disk confocal attachment was added to the left imaging port of the microscope. This combination of techniques enabled mechanical stimulation of live cells and simultaneous visualization of molecular dynamic events at the subcellular level in real-time. A PLAN APO TIRF 60x oil 1.45 NA objective lens (Olympus Corp., NY) was used for imaging live cells expressing fluorescent protein constructs excited by a Stabilite 2018 RM laser (Spectra Physics/Newport, Mountain View, CA) using a dual 488/568 nm bandpass filter from Chroma Technology (Brattleboro, Vermont). Confocal images were acquired as 3D stacks of 20 planes at a 0.25 μm step size with an exposure time of 100 ms using a QuantEM 512SC camera (Roper Scientific Photometrics, Tuscon, Arizona). The fluorescence imaging was controlled by Slidebook software (Intelligent Imaging Innovations, Denver, CO).

**AFM mechanical stimulation of VSMCs.** Tensile stress was applied to live VSMCs using an atomic force microscope probe with a 2 μm glass bead functionalized with fibronectin (Novascan Technologies, IA, USA) [9]. Formation of a functional linkage between the fibronectin on the AFM probe and cortical cytoskeleton via integrins enabled mechanical stimulation of the cell through the application of tensile forces. A mechanical stimulation experiment consists of four segments of force application. First, the probe is brought in contact with the cell for 20min to allow the formation of a functional adhesion through recruitment of integrins and focal adhesion proteins. During this time, the probe rest on the cell surface, and no tensile force is applied. The second step consists of the application of small tensile forces (i.e., mechanical stimulations of 0.2–0.4 nN) to further reinforce the adhesion by enhancing protein recruitment at the respective site. Then, the mechanical stimulation of the cell with low (~0.5 nN) and high (~1 nN) magnitude forces consisted of controlled upward movement of the cantilever in discrete steps at every 3–5 minute intervals. The same force regime mechanical stimulation was applied for 20–25 minutes each, while the actin cytoskeleton was imaged by spinning-disk confocal microscopy after each force application [9]. The AFM data were acquired using NanoScope 6.14R1 software (Veeco Instruments, Santa Barbara, CA) and were processed off-line in MATLAB (Mathworks) and Excel (Microsoft).

**Three-dimensional image analysis.** For each raw three-dimensional (3D) image volume at a specific time point, imaging data in z-direction were interpolated by linear interpolation

to generate a new sequence. Spatial sizes of a voxel in three dimensions were not all equal, i.e., $\Delta x = \Delta y = 0.178 \ \mu m$, and $\Delta z = 0.25 \ \mu m$. The resulting image sequences were imported to Imaris (v.9.3.0, Oxford Instruments, Inc.) for Automatic Tracing analysis. The coordinates of branch points from the tracing analysis was exported and saved. The 3D coordinates of all paired points that are 10 points apart along a given trace were used to compute the alignment index:

$$\text{Alignment Index} = \langle \cos\theta \rangle = \langle \frac{\Delta z}{\sqrt{\Delta x^2 + \Delta y^2 + \Delta z^2}} \rangle, \tag{1}$$

where $\Delta x$, $\Delta y$, $\Delta z$ are differences of the paired points in x, y, and z direction, respectively. The resulting set of measurements along each trace were averaged as an estimate for the angle between each trace and the z-axis. As an aggregated measure for trace angles at each time point, angle measurements from all traces at a given time point were further averaged.

## Simulation methods

A computational model for mechanochemical dynamics of active networks (MEDYAN) [41] was used to simulate the actin cytoskeletal network with an external tensile (i.e., z-axis) force. In this model, actin filaments are treated as "cylinders" connected into chains. The cylinder itself is unbendable, and the radial deformation of filaments is realized by bending between two neighboring connected cylinders. Each cylinder consists of up to 40 actin monomers, where a full cylinder is 108 nm long and has 4 possible binding sites for myosin motors and crosslinkers. Myosin motors are modeled as harmonic springs that can walk towards filament plus end with equilibrium length from 175 nm to 225 nm based on the non-muscle myosin II (NMII) [50]. Crosslinking proteins are also modeled as harmonic springs with an equilibrium length for α-actinin (30–40 nm) [51]. The main chemical events we considered in this work include filament polymerization and depolymerization, binding and unbinding of myosin and crosslinker, and myosin activation. These reactions are mechanochemically sensitive and are modeled by an efficient Next Reaction Method based on the Gillespie algorithm [52, 53]. Simulation parameters and other model details can be found in Supplementary Information and a previous publication [41].

We initialized a $3\times3\times1.25 \ \mu m^3$ simulation volume with a 250 nm radius semi-spherical AFM probe that was attached to the upper boundary. At time 0 sec, 300 seed filaments, each with 40 monomers, were randomly created in the network, defined as the free filament pool. These filaments free from simulated AFM probe attachment are allowed to polymerize and depolymerize on either the plus end or the minus end. To appropriately transmit the external force generated by probe displacement to the actin network, additional 30 seed filaments were initialized with their minus-end attached to the simulated AFM probe via stiff harmonic springs (Fig 2A). These filaments are allowed to polymerize and depolymerize at the plus end. Myosin II concentration is 2 μM (equivalent to 0.1 μM NMII mini-filament) and α-actinin concentration is 2 μM, based on their concentrations reported in *Dictyostelium discoideum* [54–56]. We use a concentration of 20 μM for actin, which is consistent with the physiological concentration of actin [57, 58]. The concentrations of actin, motors, and crosslinkers in the computational model were also used in prior computational modeling works [39, 41]. Based on an earlier work using MEDYAN, these concentrations are adequate for filament bundle to maintain their structure [39]. At the start of simulations, free G-actin was added to the network to ensure the total actin concentration is 20 μM. Since the concentration is much larger than the critical concentration [59], seed filaments would grow rapidly and reach an average F-actin length of ~0.8 μm in a few seconds of simulation. Myosin motors and α-actinin

crosslinkers were added after 5 seconds of simulation. The addition of myosin and α-actinin linkers connect the free filament pool to the filaments attached to the probe.

The external tensile force from the AFM probe was implemented as follows. The network was allowed to evolve for 150 s before the AFM probe vertical displacement (i.e., tensile force on z-axis). Each probe displacement created a 250 nm or 500 nm step displacement of the AFM probe, applying tensile force to the AFM probe-attached filaments via stiff harmonic springs. To ensure the energy was properly minimized, each displacement step was broken up into 100 sub-steps (2.5 nm or 5 nm displacement per 0.01 s). Networks were mechanically equilibrated after each sub-step, and displacement would create additional simulation space by raising the upper boundary. Since all AFM probe displacements were finished in 1s and each mechanical minimization was instant in the simulation, we were able to treat the network change before and after displacement as a fast, mechanical response that is independent of chemistry. Networks were allowed to evolve for another 150 s before the next probe pulling step (Fig 2B). During the 150 s period, cytoskeletal network remodeling was chemically dominated by filament treadmilling, myosin activation, and α-actinin linker binding and unbinding. Since the time interval between two displacement steps is much longer than the pulling time (1 s), we define the network evolution during each 150 s as the long timescale chemical response. We applied the AFM-probe pulling 5 times, for a total of 900 seconds, during each simulation. Table 1 lists all the modeling parameters.

**Table 1. Parameters for the simulations.**

| Reaction rates (unit of $s^{-1}$) | Value (reference) |
|---|---|
| Actin diffusion | 80 [41] |
| α-actinin diffusion | 8 [41] |
| Non-muscle myosin II (NMII) mini-filament diffusion | 0.8 [41] |
| Actin polymerization at plus end | 0.151 [60] |
| Actin polymerization at minus end | 0.017 [60] |
| Actin depolymerization at plus end | 1.4 [60] |
| Actin depolymerization at minus end | 0.8 [60] |
| NMII head binding | 0.2 [61] |
| NMII mini-filament unbinding under no external load | 0.2 |
| α-actinin binding | 0.009 [62] |
| α-actinin unbinding under no external load | 0.3 [62] |
| Mechanical parameters | |
| Parameters | Value |
| Length of cylindrical actin filament segment | 108 nm [63] |
| Actin filament bending energy | 672.5 pNαnm [63] |
| Actin filament stretching constant | 100 pN/nm [41] |
| Actin filament excluded volume repulsion constant | 100000 pN/nm [41] |
| NMII head stretching constant | 2.5 pN/nm [64] |
| α-actinin stretching constant | 8 pN/nm [65] |
| Boundary repulsion energy | 41 pN·nm [66] |
| Boundary repulsion screening length | 2.7 nm [66] |
| Mechanochemical parameters | |
| Force Parameters | Value |
| Unbinding force of NMII head | 12.6 pN [67] |
| Stall force of NMII head | 15 pN [41] |
| Characteristic unbinding force of α-actinin | 17.2 pN [68] |
| Characteristic polymerization force of actin filaments | 1.5 pN [69] |

The present work tested four different tensile force conditions. For convenience, we labeled them as Case i-iv in decreasing order of displacement sizes (Fig 2B). In Case i, a constant 500 nm step size was applied. This step size exerted an instantaneous force on the AFM-probe attached filaments. In Case ii, we used mixed step sizes: in the first three pulling events, each step generates 250 nm displacement, and in the last two pulling events, each step generates 500 nm displacement. In Case iii, we reduced the displacement size to constant 250 nm, implying a weaker external force. In the last case, we did not apply any external force to the network, hence, all 330 filaments were in the free filament pool. However, the upper boundary in Case iv would still move up in the same way as for Cases i to avoid any problems due to the boundary effects.

## Supporting information

**S1 Fig.** (a) The probability distribution of filament polarity alignment index for bundle-like networks under pulling condition Case i. Data are taken from t = 751s–900s out of 5 duplicated trajectories. (b) The polarity alignment index is defined as $\cos'\theta$, where $\theta'$ is the angle between a filament vector and the force direction. The filament vector (red arrow) in this case, considers the polarity of plus end and minus end. (a-b) The distribution spreads across [−1,1], suggesting that the generated actin bundles have mixed polarity.
(TIF)

**S2 Fig.** (a) F-actin radial distribution after the last pulling event (t = 751 - 900s) under pulling condition Case i-iv. (b) Representative snapshots at t = 900s for each case.
(TIF)

**S3 Fig.** (a) Representative snapshot of actin network with a static AFM probe at t = 700 s. The height of AFM probe is fixed at 1750 nm. (b) Representative snapshot of actin network with no pulling force (control case iv). (c) The alignment index for static AFM-probe (red) and no force condition (green). Error bars represent the standard deviation from the mean from 5 replicate simulations.
(TIF)

**S4 Fig.** (a) Representative snapshot of actin network with 5 filaments attached to the AFM probe, after the fifth pulling event ($d$ = 500 nm). (b) Representative snapshot of actin network with 60 filaments attached to the AFM probe, after the fourth pulling event ($d$ = 500 nm). Actin filaments, myosin motors, and crosslinkers are shown in red, blue, and green cylinders, respectively. The gray sphere represents the AFM probe.
(TIF)

**S1 Video.** Movies of VSMC expressing mRFP1-actin-7 (red) under AFM pulling, used with permission from JOVE [21].
(MP4)

**S2 Video. Actin filament bundle formation under tensile force induced by a simulated AFM-probe with step size d = 500nm (Case i pulling condition).** The network contains 330 filaments with 30 filaments attached to the simulated AFM-probe. The gray sphere represents the simulated AFM probe, and red, blue, and green cylinders represent the actin filaments, crosslinkers, and myosin motor mini filaments, respectively. $C_{actin}$ = 20 μM, $C_{myosin}$ = 2 μM, and $C_{crosslinkers}$ = 2 μM.
(MP4)

**S3 Video. Actin network geometrically contracts into cluster-like structure without external force.** The network also contains 330 filaments, and no filaments are attached to the simulated AFM probe. Red, blue, and green cylinders represent the actin filaments, crosslinkers, and myosin, respectively. $C_{actin}$ = 20 μM, $C_{myosin}$ = 2 μM, and $C_{crosslinkers}$ = 2 μM.
(MP4)

**S4 Video. Actin network evolution showing AFM-probe detachment at 600 s.** The pulling pattern is Case i (d = 500nm) at t = 150 s, 300 s, and 450 s. The 30 filaments attached to the AFM-probe were anchored to the probe until t = 600 s. At t = 601 s, the filaments detached from the probe. The gray sphere represents the simulated AFM probe, and red, blue, and green cylinders represent the actin filaments, crosslinkers, and myosin motor mini filaments, respectively. $C_{actin}$ = 20 μM, $C_{myosin}$ = 2 μM, and $C_{crosslinkers}$ = 2 μM.
(MP4)

**S5 Video. Actin network evolution under Case i pulling condition (d = 500nm), but myosin concentration is reduced to 0.4 μM.** Under this condition, the network does not contract, and the majority of the network remains random throughout the simulation. The gray sphere represents the simulated AFM probe, and red, blue, and green cylinders represent the actin filaments, crosslinkers, and myosin, respectively. $C_{actin}$ = 20 μM, and $C_{crosslinkers}$ = 2 μM.
(MP4)

**S6 Video. Actin network evolution under Case i pulling condition (d = 500nm) with lower crosslinker concentration ($C_{crosslinkers}$ = 0.4 μM).** Although the network still contracts, the filaments attached to the AFM probe disconnected from the free actin filament pool after ~ 300s. Eventually, the networks become a small filament bundle attached to the AFM probe at the top of the network and a disconnected larger cluster at the bottom. The gray sphere represents the simulated AFM probe, and red, blue, and green cylinders represent the actin filaments, crosslinkers, and myosin, respectively. $C_{actin}$ = 20 μM, and $C_{myosin}$ = 2 μM.
(MP4)

**S7 Video. Actin network evolution under d = 500 nm tensile displacement size with the time interval between two displacements reduced from 150s to 10s.** The network is first allowed to evolve for 160s before the first pulling event. The video shows the trajectory between 130 ~ 198s with four pulling events in total. The gray sphere represents the simulated AFM-probe, and red, blue, and green cylinders represent the actin filaments, crosslinkers, and myosin, respectively. $C_{actin}$ = 20 μM, $C_{myosin}$ = 2 μM, and $C_{crosslinkers}$ = 2 μM.
(MP4)

## Acknowledgments

We would like to thank Aravind Chandrasekaran and Carlos Floyd for insightful discussions. MEDYAN simulations were carried out on the HPC (High Performance Computing) resources at Georgia State University and Deepthought2 Supercomputers at the University of Maryland. The authors also thank Dr. Amanda Howard for data visualization in Imaris software (Bitplane, Inc.) for Fig 1B–1C, right panels.

## Author Contributions

**Conceptualization:** Andreea Trache, Yi Jiang.

**Data curation:** Soon-Mi Lim, Jerome P. Trzeciakowski, Andreea Trache.

**Formal analysis:** Xiaona Li, Qin Ni, Jun Kong.

**Funding acquisition:** Andreea Trache, Yi Jiang.

**Investigation:** Xiaona Li, Qin Ni, Xiuxiu He, Soon-Mi Lim, Andreea Trache, Yi Jiang.

**Methodology:** Qin Ni, Jun Kong, Andreea Trache.

**Project administration:** Yi Jiang.

**Resources:** Yi Jiang.

**Software:** Qin Ni, Garegin A. Papoian.

**Supervision:** Garegin A. Papoian, Andreea Trache, Yi Jiang.

**Validation:** Andreea Trache, Yi Jiang.

**Visualization:** Andreea Trache, Yi Jiang.

**Writing – original draft:** Xiaona Li, Qin Ni, Yi Jiang.

**Writing – review & editing:** Qin Ni, Xiuxiu He, Garegin A. Papoian, Andreea Trache, Yi Jiang.

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
