## [Decision Letter · Decision Letter 0]

3 Mar 2020

Dear Dr. Jiang,

Thank you very much for submitting your manuscript "Tensile Force Induced Cytoskeletal Reorganization: Mechanics Before Chemistry" for consideration at PLOS Computational Biology.

As with all papers reviewed by the journal, your manuscript was reviewed by members of the editorial board and by several independent reviewers. In light of the reviews (below this email), we would like to invite the resubmission of a significantly-revised version that takes into account the reviewers' comments.

We cannot make any decision about publication until we have seen the revised manuscript and your response to the reviewers' comments. Your revised manuscript is also likely to be sent to reviewers for further evaluation.

Sincerely,

Jeffrey J. Saucerman

Associate Editor

PLOS Computational Biology

Daniel Beard

Deputy Editor

PLOS Computational Biology

Reviewer's Responses to Questions

**Comments to the Authors:**

Reviewer #1: This manuscript presents important and interesting findings from molecular simulations and experiments about dynamic remodeling of actin network under the influence of external force. Both experiments and simulations demonstrated fast actin alignment to external force followed by slow myosin-mediated stabilization of the bundled actin network, termed as “mechanics before chemistry” by the authors. The videos of the simulated dynamic remodeling of actin networks under various conditions are particularly enlightening. Findings from this work has strong implications for mechano-transduction and stress fiber formation in cells. The paper is clearly written.

I have the following comments/questions about the manuscript.

1.Comparing the default case and the case with 10 s between pulling, it seems like a proper balance between the pulling rate and the biochemical relaxation rate is important for maintaining a continuous actin bundle. Is dynamic pulling necessary though? Could the actin network remodeling be induced just by geometry? For example, if the AFM probe is initially placed 2500 nm away from the initial random actin network and stay there, would an actin bundle similar to that in case iv form?

2.Seems like Case iv without pulling was simulated without the 30 AFM-attached actin filaments. Addition of the AFM-attached filaments alone (without pulling) should induce actin alignment, which explains why in Fig.3c case i-iii has slightly higher alignment index than case iv without pulling in the first 150 seconds. This effect is very weak though, probably because the AFM probe is initially right on top of the initial random actin network. Related to the question above, if the AFM probe is initially set farther away from the initial actin network, would the above effect become stronger?

3.“With a value between 0 and 1, the alignment index equals to 1 for perfect alignment with the Z-axis, 0.5 for completely random, and 0 for alignment perpendicular to the Z-axis.” According to the definition of the alignment index, average of cos(theta), completely random directions of actin between 0 and pi/2 should give an alignment index of 2/pi = 0.64. Why would it be 0.5 in this case? It is also noted that in Fig. 3c, case iv, the alignment index initializes around 0.5 and gradually drops to around 0.4. Why would the index drop to 0.4 for globular actin cluster?

Typos found:

1.“The alignment index increases immediately after each of the AFM pulling events in all three pulling patterns tested (Figure 4c, Case i-iii).” Should be Figure 3c.

2.“However, the upper boundary in Case iv would still move up in the same way as for Cases i-iii to avoid any influences from the boundary effects.” Case i-iii move in different ways. Case iv should move up in the same way as for Cases i, as stated in the caption of Fig. 2b.

Reviewer #2: In their manuscript, Li and Ni et al. investigate a fundamental question in the cellular mechanotransduction field: how do cells sense and react to externally applied mechanical stimuli? To do this, they combine experiments performed in living cells with computational modeling performed in silico to investigate the ability of the actin cytoskeleton within a cell to dynamically respond to tensile external force. Specifically, their experiments consisted of imaging and quantifying the reorganization of the actin cytoskeleton within living vascular smooth muscle cells (VSMCs) expressing mRFP1-tagged actin as an external tensile force was applied to the their apical plasma membrane via a fibronectin-functionalized atomic force microscopy (AFM) probe. They then used computational modeling to simulate the mechanical structural adaptation of active actin cytoskeleton networks in response to externally applied tensile force in a manner similar to their AFM-based VSMC experiments. Based on the results of their experiments and modeling, the authors conclude that mechanical structural adaptation occurs before chemical adaptation during actin bundle formation. Basically, Li and Ni propose that actin filaments first align in the direction of the externally applied tensile force, which results in their anisotropic orientation followed by the chemical evolution of the actin network into a dense bundle-like geometry. They call this a “mechanics before chemistry” model of actin cytoskeleton remodeling in response to externally applied force. While I feel that this manuscript tackles an important and timely question in the field of cellular mechanobiology, I have several major and minor issues that I strongly feel that the authors need to address before I can recommend this work for publication at PLoS Computational Biology. In addition, there were several grammatical/writing issues that I would like the authors to address. These issues are all described below.

Major Issues:

1)The authors need to do a better job of motivating their work. Why is it important that they “ask how tensile force induces cytoskeletal remodeling and the active formation of actin bundles” in VSMCs?

2)Where is the AFM probe placed on the apical surface of the VSMCs? Is it placed in the same relative position? Is it place on or away from the nucleus? More information is needed.

3)The authors need to explain why they chose the specific concentrations of the molecules included in their computational models. What are the references that these concentrations were pulled from and in which cell types?

4)Why were only 30 filaments attached to the AFM tip in the simulations shown in Fig. 2? What happens if the number of attached filaments is increased or decreased? Are the filaments always attached to the AFM tip or do the have a dissociation constant?

5)The simulation described in Fig. 2A does not seem to be physiologically relevant to the AFM experiments performed on live VSMCs shown in Fig. 1. I say this because the authors are measuring the axial displacement of fluorescently labeled pre-assembled actin stress fibers caused by the pulling of an AFM tip on the apical surface of a VSMC, while the simulations are pulling on an active network of actin filaments that evolves over the time that tensile force is applied to the network. In addition, the geometry of the stress fibers in the VSMCs is quite noticeably different from the geometry of the simulated AFM experiments. Were the authors to have measured the accumulation of an active cytoskeletal network underneath the adhered AFM as it was pulled away from the apical VSMC plasma membrane, the simulations presented in this work would be more directly comparable.

6)I cannot agree with the authors’ description of their simulated active actin networks under tension as being stress fiber-like. While the simulated networks and stress fibers both have mixed filament polarity, the architecture of these two actin structures are quite different. The architecture of a stress fiber is reminiscent of a contractile sarcomere in muscle, which are “blocks of actin filaments of alternating polarity and bands of interdigitating non-muscle myosin” (Pellegrin and Mellor, 2007 J Cell Science). Pellegrin and Mellor go on to say, “the structure (i.e. sarcomere /stress fiber) is held together by α-actinin and also crosslinked by non-muscle myosin”. In contrast, the authors’ simulated active actin networks more closely resemble the actin “comet tails” responsible for the intracellular movement of endosomes, some bacteria (i.e. Listeria monocytogenes and Rockettsia conorii), and some viruses (i.e. Vaccinia and Baculoviruses) (Welch and Way, 2013 Cell Host Microbe; Fehrenbacher et al., 2003 J Exp Biol). Given this disconnect between the VSMC experiments and the simulations, I find it difficult to see how the in silico modeling complements or illuminates the in vitro cellular experiments presented in this work.

7)I would very much like to seem movies of the AFM experiments performed on VSMCs expressing the so-called actin-mRFP1 construct (see Minor Issue #5 below).

8)In lines 5-6 of the 2nd paragraph of the “Two-step development of actin bundles relies on both faster mechanical alignment and slower chemical stabilization” section of the Results, the authors state, “The control case without external pulling (iv, green line) shows the biochemically driven F-actin accumulation, as a result of myosin-induced contractility”. Why do they authors think that this is all due to myosin-induced contractility and not also due to alpha-actinin-induced bundling and/or actin polymerization?

9)Why do the authors only include “4 possible binding sites for myosin motors and crosslinkers” for a 40 subunit 108 nm long cylinder of an actin filament? This is a gross underestimation, as a myosin-II motor can interact with each actin subunit in a filament. Therefore, I would assume that there were 40 myosin-binding sites per actin filament cylinder. What happens if the number of binding sites were increased in the computational models?

10)I would recommend that the authors tone down their conclusion that their “mechanics before chemistry” hypothesis is novel. It is well known in mechanobiology that the timescales of mechanotransduction range from milliseconds to seconds for the stretching of mechanosensor proteins to the polymerization of actin, respectively.

Minor Issues:

1)In the legend of Fig. 2, the authors need to state how long the 300 free actin filaments are that are in their simulations.

2)In the 5th line of the 4th paragraph of the Discussion, the authors might elaborate on which specific signaling pathways might be triggered by stretching focal adhesion proteins.

3)In the last paragraph of the Discussion, the authors state, “we integrated in vitro and in silico modeling”. However, I do not understand what is meant by “in vitro modeling” in this statement. It seems to me that the authors integrated cellular biophysical experiments with in silico modeling instead.

4)In the last sentence of the Discussion, the authors state, “This result suggests….which can have important implications to mechano-signal transduction. Perhaps they might provide some examples of such “important implications”? This would give the reader a sense of where the authors plan on going next with this line of research.

5)The authors state that they are expressing an actin-mRFP construct that they received from the late Michael Davidson in VSMCs to visualize the actin cytoskeleton. However, a quick search of Michael’s fluorescent protein construct collection for a red actin construct suitable for mammalian expression revealed that he had a construct named “mRFP1-actin-7”, not one named “actin-mRFP”. Therefore, the authors should change all mentions of “actin-mRFP” to “mRFP1-actin-7”. This change is important for others to be able to reproduce the authors’ results. Moreover, the way that the authors write “actin-mRFP” suggests that the mRFP is fused to the C-terminus of actin, which is not correct.

6)In the Materials and Methods, the authors cite direct their readers to reference #45 for detailed information regarding the “integrated microscope system” used in this manuscript. While I understand that they do not want to have to go into all of the details of their previously described imaging system, I do feel that they need to at least provide their readers with the following information:

a.The name of the microscope company that the base of the microscope was built on.

b.The numerical aperture value for their 60x objective and the name of the microscope company that manufactured it.

c.The cameras used and the companies that made them.

d.The light source (I assume that they are lasers, but which ones and from where?).

e.The information about the excitation and emission filters used.

f.The software used to drive the imaging system.

7)What was the concentration of fibronectin used to coat the AFM probe?

8)In the “Simulation Methods” section of the Methods, I do not understand what the authors mean by the phrase “strong axial stretching stiffness”.

9)In the “Simulations Methods” section of the Methods, the authors need a reference for the following statements:

a.“Myosin motors are modeled as harmonic springs…pm the non-muscle myosin II”.

b.“Crosslinking proteins are also modeled as harmonic springs…(30-40 nm)”.

Writing Suggestions:

1)In the 3rd line of the 2nd paragraph of the Introduction, the authors state, “is critical in understanding”. They should change the “is” to a “for”.

2)In the 6th line of the 4th paragraph of the Introduction, the authors should change “myosin activation” to ”myosin-II activation”.

3)In the 1st line of the 5th paragraph of the Introduction, the authors should remove the “the” following “upon”.

4)In the 3rd line of the 5th paragraph of the Introduction, the authors should insert the word “software” following “Network)”.

5)In the heading of the 1st section of the Results, the authors should change the “of” before “live” to “in”.

6)In the 1st line of the 2nd paragraph of the “Two-step development of actin bundles relies on both faster mechanical alignment and slower chemical stabilization” section of the Results, the authors need to change “filament” to “filaments” and “keeps” to “kept”.

7)In the 3rd line of the 2nd paragraph of the “Two-step development of actin bundles relies on both faster mechanical alignment and slower chemical stabilization” section of the Results, the authors need to change “calculate” to “calculated”.

8)In the 5th line of the 3rd paragraph of the Discussion, the authors should change “isotropically” to “isotropic”.

9)In the 2nd line of the last paragraph of the Discussion, the authors should insert “actin” before “cytoskeleton”.

10)In the last sentence of the Discussion, the authors should change the “to” to a “for”.

11)In the 2nd to last sentence of the “Experimental Methods” section of the Methods, the authors should change the “of” before “Michael” to “from”.

12)In the “AFM mechanical stimulation to VSMCs” section of the Methods, the authors should make the following changes:

a.Change the “to” before “VSMCs” to “of” in the heading of this section.

b.Change the two “consists in” to “consists of”.

13)In the “Three-dimensional image analysis” section of the Methods, the authors should make the following change:

a.In the penultimate sentence before the Alignment Index equation, the word “was” needs to be changed to “were”.

14)In the “Simulations Methods” section of the Methods, the authors need to make the following changes:

a.In the 2nd paragraph, the “allowing” found in the sentence that begins with “Only plus ends of these filaments” needs to be changed to “allowed”.

b.In the 3rd paragraph, the “tips” in the sentence that begins with “Each probe displacement” needs to be changed to “tip” and the “generating” needs to be changed to “applying”.

c.In the 3rd paragraph, the “are” in the sentence that begins with “Since all AFM” needs to be changed to “were”.

15)In the legend of Fig. 1, the authors need to delete the “were” from the description of panel (a).

16)In the legend of Fig. S1, the “is” in the sentence that begins with “The distribution spreads” needs to be changed to “are” and ”a” needs to be inserted before “stress”. In addition, a reference for this statement needs to be provided.

**Have all data underlying the figures and results presented in the manuscript been provided?**

Reviewer #1: Yes

Reviewer #2: Yes

PLOS authors have the option to publish the peer review history of their article (what does this mean?). If published, this will include your full peer review and any attached files.

Reviewer #1: No

Reviewer #2: No
---

## [Decision Letter · Decision Letter 1]

21 Apr 2020

Dear Dr. Jiang,

We are pleased to inform you that your manuscript 'Tensile Force-Induced Cytoskeletal Remodeling: Mechanics Before Chemistry' has been provisionally accepted for publication in PLOS Computational Biology.

Best regards,

Jeffrey J. Saucerman

Associate Editor

PLOS Computational Biology

Daniel Beard

Deputy Editor

PLOS Computational Biology

Reviewer's Responses to Questions

**Comments to the Authors:**

Reviewer #1: The authors have fully addressed all my comments.

Reviewer #2: I thank the authors for having addressed all of my concerns, especially for improving the clarity of their manuscript. I feel that the manuscript is acceptable for publication.

**Have all data underlying the figures and results presented in the manuscript been provided?**

Reviewer #1: None

Reviewer #2: Yes

PLOS authors have the option to publish the peer review history of their article (what does this mean?). If published, this will include your full peer review and any attached files.

Reviewer #1: No

Reviewer #2: No

---

## [Editor Report · Acceptance letter]

3 Jun 2020

PCOMPBIOL-D-20-00147R1 

Tensile Force-Induced Cytoskeletal Remodeling: Mechanics Before Chemistry

Dear Dr Jiang,

I am pleased to inform you that your manuscript has been formally accepted for publication in PLOS Computational Biology. Your manuscript is now with our production department and you will be notified of the publication date in due course.

With kind regards,

Laura Mallard
